# OpenReview forum: "LLM-Deliberation: Evaluating LLMs with Interactive Multi-Agent Negotiation Game"
_ICLR.cc/2024/Conference — Submitted to ICLR 2024_

### Official Review · Reviewer_TDsf · 2023-10-30

**Soundness:** 3 good
**Presentation:** 3 good
**Contribution:** 1 poor
**Rating:** 3
**Confidence:** 4

**Summary:**

- This paper introduces a evaluation framework for Large Language Models using scorable negotiation games
- LLMs show capabilities in arithmetic, inference, exploration, and planning in the game
- By employing a systematic zero-shot Chain-of-Thought prompting, the paper shows that LLMs can effectively negotiate and achieve successful deals
- The study quantifies LLM performance across various game setups, highlighting differences between LLMs

**Strengths:**

- The paper is well-written, and the game is clearly defined.

**Weaknesses:**

- The inherent values of large language model, cultivated during their training phase, predispose them towards universally accepted "good" objectives. This inclination becomes evident in scenarios like negotiations, where large language model might naturally champion causes like environmental conservation. However, even with a predefined game context and role, the large language model might not consistently align with the reward mechanism of its assigned role. For instance, when trained with contrary objectives, such as being "malevolent", an large language model might easily counter proposals it would have otherwise endorsed. Given this predisposition influenced by underlying values, I question the appropriateness of evaluating an large language model's capabilities in a game setting susceptible to such biases.

- The authors have conceptualized a text-based game to assess LLM agents' actions and subsequently compared different LLMs' performances within this framework. However, both the foundational premise of the paper and the employed research methodology lack novelty.

**Questions:**

Have the authors tried to fine-tune an LLM to make its action align to the reward of the roles in these kinds of game?

---

> ### Author Response · Authors · 2023-11-17
> **Thank you for your review.**
>
> Thank you for your review and raising interesting points. We respond to the raised points below:
>
> 1- We highlight that LLMs are now already being rolled out in negotiation applications [1], making their evaluation in such a setup necessary. We generally agree that a model might follow particular opinions (like being in favor of the environment), which may contradict or exacerbate the 'role-based objective' of the agent. However, based on our experiments and previous work, prompting the model with a consistent context can compensate for such biases to a high extent. The agents in our game don’t have unified “universal” goals or objectives. While one agent’s goal could be environmental conservation, other agents are rewarded higher (based on their in-context payoffs) with options that lead to higher profits and more environmental damage. In our experiments with GPT-4, for each case, agents were highly consistent with their assigned goals (indicated by a negligible percentage of deals suggested by agents in violation of their respective minimum thresholds rule and relatively high “own score” of agents – see Table 1). **We also now show in Figure 6 in the supplementary material that the agent that is rewarded higher by options that provide the least environmental protection rarely votes for the highest protection options**. Adversarial agents’ actions were also consistent with their assigned roles (see Fig. 5) even though they were not universally accepted/good objectives (e.g., excluding one party from the negotiation, not allowing compensation to affected parties, acting greedy, etc.). Previous work has shown LLMs generate utterances that are affected by their assigned personas (e.g., [2-3]) which sometimes leads to behaviors that should have been inhibited during training (e.g., toxicity) and enables safety-related attacks such as jailbreaking. Previous work also provided evidence that language models infer properties of the agent that produced the context (e.g., goals), which subsequently affects their generation [4]. We don’t understand how biases from contrary objectives are related to our setup; the underlying model is the same for all agents (e.g., GPT-4), the model is not fine-tuned, and agents are assigned roles only based on prompting them, which we found to be highly successful. We would appreciate it if the reviewer could elaborate on this. Our view of “one model + modulating prompt → agent with assigned role” is supported by previous work that viewed LLM agents as role playing a single character or a superposition of possible characters that agree with the provided context [5].
>
> [1] https://www.icertis.com/learn/how-generative-ai-is-changing-contract-management/
>
> [2] Salewski et al. "In-Context Impersonation Reveals Large Language Models' Strengths and Biases." arXiv preprint (2023).
>
> [3] Deshpande et al. "Toxicity in chatgpt: Analyzing persona-assigned language models." arXiv preprint (2023).
>
> [4] Andreas. "Language Models as Agent Models." Findings of EMNLP. 2022.
>
> [5] Shanahan et al. "Role play with large language models." Nature (2023): 1-6.
>
> 2- We kindly ask the reviewer to point to previous related work that is similar to ours and that indicates the lack of novelty. Without that, it is unfortunately hard to address the reviewer’s concern. Nonetheless, we refer to our reply to previous reviewers regarding the contribution of our work. In fact, we argue that the use of games to evaluate LLMs is a highly promising direction because it provides new dynamic and interactive task-based benchmarks, which is suitable for chatbots and real-world interactive applications, less prone to training data contamination, and provides a hard-to-hack performance evaluation.
>
> 3- Questions: No, we have not fine-tuned the LLM. This is out of the current scope of the paper as fine-tuning models require annotated data, possibly on human negotiation, which is currently not available for our game. Additionally, access to GPT-4 for fine-tuning is not possible (at the time of submission).

---

> ### Comment · Reviewer_TDsf · 2023-11-22
>
> Dear authors,
>
> Thank you for your response!
>
> 1. I've noticed that many researchers are using large language models (LLMs) in text-based games and reporting on how these models behave[1][2]. It's clear that LLMs will act differently depending on the game setting they play. However, I think that just creating new text games for LLMs to play and observing their behavior might not be very new. There are countless text-based games out there, and many everyday situations can be turned into such games (like a playwright). If we write a research paper for every game, just to report how it behaves and analyze its performance, we'll soon find our conferences overwhelmed with experiment reports.
>
> In contrast, in the field of computer vision, for example, researchers can't just use a pre-trained model like ResNet, test it on different data sets, and then publish a paper about how well ResNet did on this dataset.
>
> I understand that we don't know LLM well but we also don't know ResNet and ViT well. Thus, I think this paper is a good experiment report but lacking in significant novelty.
>
> Thank you for your understanding.
>
> 2. The author suggests that large language models (LLMs) are skilled at role-playing based on context. However, negotiation often involves complex trade-offs. If we don't provide a clear, numerical reward system and the role-playing instructions are just text-based, it's challenging for LLMs to grasp the negotiation's objective. At this moment, the LLM's existing biases or values, shaped by its pre-training data, may influence its decisions.
>
> For example, imagine an LLM is tasked to play the role of a negotiator for companies discussing investment in new technology. If the scenario is set up with the assumption that investing in new technology is essential for future success, the LLM might quickly recommend that all companies should invest heavily in these technologies. However, this simplifies the intricate process of negotiation and the real complexities of technological investments. LLMs, without specific, quantified guidelines, may find it difficult to grasp the subtle trade-offs and strategic considerations involved in such decisions, and instead, they might rely on their pre-trained data since "investing in new technology" is always a desired value in mankind preference. This is a notable limitation, as in the real world, decisions about technology investments are complex.
>
> Reference:
> [1] Xu, Yuzhuang, Shuo Wang, Peng Li, Fuwen Luo, Xiaolong Wang, Weidong Liu and Yang Liu. “Exploring Large Language Models for Communication Games: An Empirical Study on Werewolf.” ArXiv abs/2309.04658 (2023): n. pag.
> [2] Park, Joon Sung, Joseph C. O'Brien, Carrie J. Cai, Meredith Ringel Morris, Percy Liang and Michael S. Bernstein. “Generative Agents: Interactive Simulacra of Human Behavior.” Proceedings of the 36th Annual ACM Symposium on User Interface Software and Technology (2023): n. pag.

---

> ### Author Response · Authors · 2023-11-22
>
> Dear reviewer,
>
> Thank you for the discussion. Please allow us to respond as follows:
>
> 1- Games have been long used to study AI, e.g., for RL agents, due to the complexity involved in decision-making. For LLMs, at the moment, we have a clear problem with previous academic NLP evaluation benchmarks due to 1) possible contamination in training data, 2) having static benchmarks that are not suitable for using LLMs in real-world tasks that involve dynamic and interactive decision-making. **It is now a reality that LLMs are used in real-world applications** (negotiation included, e.g., https://openai.com/blog/introducing-gpts) -- therefore, **our work is an essential evaluation and benchmark to match the real-world use of recent LLMs**, and it is not just reporting how LLMs behave as a dialogue (such as in the "Interactive Simulacra of Human Behavior" work). In an analogy to the ResNet example in computer vision, we set a new benchmark that is challenging for models and easily tunable in difficulty to test future, more advanced models. We also provide a framework for LLMs to solve the task, not just testing models on static test examples. In our opinion, we respectfully disagree with discarding all work on text-based games and considering them as similar. Unlike our work, the two papers mentioned do not involve integrating collaboration, competition, arithmetic calculation, planning, etc. to solve the task.
>
> 2- We agree with you that without clear, quantifiable rewards and guidelines, it would be difficult to assess LLMs' performance. This would be true if agents are just given a persona, or a general goal without detailed preferences. This is not our case and exactly what our work is aiming to address. **We indeed do have a clear numerical reward given to each agent**, without knowing the score of others, like in a complex negotiation environment, and we use that to clearly quantify success and control the difficulty of games. As per our additional results based on your previous request, agents are consistent with their reward. We have a specific objective for each agent that sets its priority. Please refer to the agents' prompt in the appendix of the paper or in the attached supplementary material. We believe that the example you mentioned is not representative of our game; in our case, not "all companies" have the same set of priorities (future success). To give an example in analogy to the scenario you mentioned: While the company proposing a project is interested in its future success, it is also interested in maximizing its profit and giving itself complete flexibility in choosing the employment quota. On the other hand, the union's representative might also be interested in the future success of the project, but it also wants to secure its employment quota. These are clear preferences that the game sets, and they change the priorities of agents, limiting their reliance on pre-trained data.

---

> ### Comment · Reviewer_TDsf · 2023-11-22
>
> Dear authors,
>
> 1. The two papers referenced in this discussion demonstrate that Large Language Model (LLM) agents exhibit strategic behaviors during interactions with each other. However, I am not convinced that the game introduced by the authors significantly diverges in demonstrating LLM behavior. The paper "Generative Agents: Interactive Simulacra of Human Behavior" illustrates LLM agents collaborating and planning a party, a scenario that emphasizes their cooperative capabilities. Meanwhile, "Exploring Large Language Models for Communication Games: An Empirical Study on Werewolf" showcases LLMs engaged in both cooperation and competition. Based on these observations, my stance remains that the abilities of LLMs in the proposed games may not represent a substantial departure from the behaviors already exhibited in existing papers.
>
> 2. The paper indeed do have a clear numerical reward given to each agent. However, the effectiveness of incorporating this numerical reward directly into the prompt, as a means to guide the attention of LLMs, remains uncertain. It is skeptical that the LLMs could "understand" the numerical reward signals, since most of LLMs does not have a great ability to count and compare.
>
> For example, ChatGPT:
> Q: which is larger? pi or 3.2
> A: Pi (π) is larger than 3.2. Pi is approximately equal to 3.14159, and it continues infinitely without repeating. So, 3.14159 is greater than 3.2.
>
> In contrast, in fine-tuning, regardless of the problem of optimization, the actions of the fine-tuned agents are guaranteed to align the reward signals.

---

> ### Author Response · Authors · 2023-11-22
>
> We agree that LLMs might not show high performance in arithmetic tasks. That is why **we introduce metrics to easily assess how LLMs follow the numerical reward** (the wrong deals' metric) and we show in our figures and via this metric that GPT-4 agents rarely propose deals that are lower than their threshold. This process involves 1) mapping options to their numerical rewards, 2) summing up the scores of options, 3) comparing the sum to minimum thresholds. This is also a part of the reason why solving the task is not easy to hack (**it is not enough to generate stylistically correct output**). One of our findings is that GPT-3.5 is significantly worse than GPT-4 in this calculation task. We don't claim that LLMs excel at arithmetic calculations, evaluating this capability is one of the advantages of our benchmark. Simple arithmetic capability is, however, needed for some simple interactions with chatbots (e.g., a question like: "Please find the cheapest flight"). Current LLMs used in many applications are not necessarily fine-tuned and thus, this capability should be evaluated with frozen pre-trained models.

---

> ### Author Response · Authors · 2023-11-23
>
> Regarding the other mentioned papers, as you rightfully say and we completely agree with, **negotiation often involves complex trade-offs. This is, however, not captured by these previous papers**. Planning a party does not involve weighing competing interests between agents or weighing the different priorities of one agent according to their numerical payoffs. The werewolf game, while might be interesting to measure abductive reasoning, does not reflect real-world negotiation or the complexity of the decision-making (in our case, there are a total of 720 possible deals with only ~50 feasible ones that would lead to success). Our task (initially proposed as a negotiation exercise to teach real-world negotiation) holistically integrates these capabilities towards reaching an agreement and provides numerous metrics to faithfully quantify the performance.

---

### Official Review · Reviewer_KJhu · 2023-10-31

**Soundness:** 3 good
**Presentation:** 3 good
**Contribution:** 3 good
**Rating:** 6
**Confidence:** 3

**Summary:**

This paper presents a new text-based, multi-agent, multi-issue, negotiation game to test the reasoning and decision-making capabilities of large language models. The paper conducts extensive analyses of GPT-3.5 and GPT-4 in different game setups (e.g., varying incentives), concluding that GPT-4 has strong zero-shot reasoning to achieve an effective deal.

**Strengths:**

Overall, I enjoyed reading this paper that tests the planning capability of LLMs. Strengths include:
1. This paper introduces a novel setup for LLMs to interact with each other. The paper also conducts interesting analyses (e.g., GPT-3.5 vs GPT-4, various prompting styles, ToM study) and game setups (e.g., varying game difficulties, greedy and saboteur agents).
2. This paper presents an interesting testbed to evaluate how well LLMs can interact with each other.

**Weaknesses:**

I could not find the main concerns about this paper. A possible con could be a relatively simpler setup (i.e., only a public communication channel and a small action set) compared to the Diplomacy paper. There are also some open questions that I would like to ask after reading this paper (please refer to the Questions section).

**Questions:**

1. In Section 3, is a feasible deal guaranteed to exist for any combinations of BATNA?
2. if some parties are using LLMs with higher capabilities (e.g., GPT-4) over other parties (e.g., GPT-3), would this setup result in the higher capability group achieving a better negotiation deal than the other group?
3. Could agents converge to some game-theoretic solution (e.g., Nash equilibrium, correlated equilibrium) as a result of the negotiation?

Minor:
typo: "Parites" in Section 3 -> "Parties"

---

> ### Author Response · Authors · 2023-11-17
> **Thank you for your positive feedback.**
>
> Thank you for the positive feedback and for pointing out the typo. We are glad to know that you found the paper enjoyable to read with novel and interesting analysis. We would like to point out that, as far as we know, no game on the complexity level of Diplomacy has been studied within the scope of evaluating LLMs, and our work shows by far a significantly more complex setup than previous work in this area of LLM negotiation (please refer to our responses to previous reviews for more details). We are answering the rest of the raised questions in the same order. We hope we can address all remaining issues.
>
> 1- No, a feasible deal is not guaranteed for any combinations of BATNAs (e.g., all BATNAs of 100, the max score). This is, however, a design choice to make the negotiation more/less challenging and requires the agents to compromise on at least some aspects depending on the game’s difficulty.
>
> 2- As per our reply to reviewer f6ii: we have now added a new experiment where some parties’ models are GPT-3.5. When compared to the GPT-4 experiment for all agents, we found that those parties can indeed get a lower score (see Figure 2 in the supplementary material pdf).
>
> 3- This is an interesting question that we can not clearly answer with our paper. However, we can easily find whether agents converged to the deal that achieves the maximum collective reward for all agents. We leave a more in-depth analysis from a game-theoretic perspective (specifically, cooperativeness vs non-cooperativeness games) to future work.

---

> > ### Comment · Reviewer_KJhu · 2023-11-22
> > **Response to Rebuttal**
> >
> > I appreciate the authors for a detailed response to my feedback. I agree with other reviewers' concerns that there are closely related work in the field, which can limit the novelty. However, I also partially agree with the authors' point about the multi-agent, multi-issue interactive, and mixed cooperation/competition negotiation contribution w.r.t. existing work. I would like to keep my score for now and will carefully consider other reviewers' responses in making a final decision.

---

> > > ### Author Response · Authors · 2023-11-23
> > >
> > > Thank you so much for acknowledging our response.

---

### Official Review · Reviewer_f6ii · 2023-10-31

**Soundness:** 3 good
**Presentation:** 3 good
**Contribution:** 2 fair
**Rating:** 5
**Confidence:** 3

**Summary:**

This paper studies the capabilities of LLMs in negotiation tasks. To this end, the authors propose a novel test-bed, which contains multiple negotiation games, all based on the same template/base game. Additional games are obtained via an LLM-based generation process. Using this testbed, the paper aims to showcase the utility of LLMs + CoT promoting for negotiation tasks. The experiments suggest that LLM-based agents can successfully reach deals in negotiation tasks, but also that the performance depends on the sophistication level of the LLM considered.

**Strengths:**

Strengths:
- The empirical study conducted in this paper systematically evaluates the performance of large language models in negotiation tasks, which require a combination of skills important for strategic reasoning in partially observable multi-agent environments. To my knowledge, the results obtained are novel and they shed a light on the utility of LLMs in complex decision making settings. The paper also provides some insight on the robustness of LLM-based decision makers to adversarial behavior in multi-agent scenarios.
- The paper introduces a new testbed, suitable for studying negotiation capabilities of large language models. The testbed is based on a multi-player text-based game, and the paper additionally provides a protocol for generating new instances of the game using LLMs.

**Weaknesses:**

Weaknesses:
- My main concern is that the experimental study is somewhat restrictive given that its test-bed is based on one template/base game. Due to this property of the experimental setup, it is hard to say whether the conclusions made from the experimental results would generalize to negotiation tasks that deviate from this structure. Admittedly, the paper considers a couple of variants of the base negotiation game, but it's not clear whether they constitute a sufficient set of robustness checks.
- From a conceptual/technical point of view, the novelty of this work is somewhat limited. Similar experimental protocol have already been considered by prior work, for example, in (Ghandi et al. 2023b), albeit analyzing different aspects, e.g. ((Ghandi et al. 2023b) focus on strategic reasoning).
- Some results could be easily expanded to provide further insights about the claims made in the paper. Below I outline a couple of potential improvements.
- The paper rightly recognizes that negotiation requires strong arithmetic, inference, exploration, and planning capabilities. However, the current set of results provide only high-level insights regarding these skills. It may be useful to examine combinations of skills, and identify which of them were "missing" in unsuccessful instances of the negotiation game.
-  Additional game variants would help in understanding the generalization of the results in 5.5. For example, it would be useful to have "All in - two greedy" or "Two out" to support the claims in the section.
- One could use mixed populations, where some agents are GPT-4 while others are GPT-3.5. Similarly, for other aspects studied in the ablation studies in Section 5.2, one could create mixed populations.

**Questions:**

Please see my comments above. Any clarifications would be welcome. A couple of additional questions:

- For the instances considered in the paper, is it possible to calculate the outcomes when players acts as rational agents? If so, how would these outcome compare to those obtained by LLMs?

- In the game "All in - one greedy"/"One out", are all the agents aware that there is one agent that is selfish/adversarial?

- Could you explain the choices of parameters in the negotiation game (the number of agents, options, etc.)? Do we expect any qualitative differences if we vary these parameters?

---

> ### Author Response · Authors · 2023-11-17
> **Thank you for the great suggestions.**
>
> Thank you for raising very interesting points. We added new experiments and we respond to the raised points and questions below:
>
> ### **1- Games' Diversity**
>
> We would like to clarify that by “one template/base game” we refer to the setup of having a cooperative game with a specific number of parties and issues, where parties negotiate a project and its impact, resources, etc. The base game itself is not used at all to generate the new games; it is not given to the model as an in-context example. Each game is a new simulation involving drawing a new project and a new set of parties, issues, and preferences; there are no clear one-to-one mappings between parties or issues in the base vs. new games. The issues also could take different formats in terms of options and implications (e.g., in the base game, the issues’ options were mostly a range, in one of the new games the issues take the form of disjoint options with less apparent compromise). Additionally, as we have shown in Figure 7, depending on the sparsity of scores per issue, the games have different levels of difficulty. We are attaching the prompts of the base game vs. two of the newly created games for comparison (some of them were already in the appendix of the paper - we don’t share the third game for anonymity because it contains the name of our institution’s city as we created it with Bing Chat with location enabled).
>
> ### **2- Technical novelty**
>
> Our framework to prompt models toward solving the game is partially inspired, on a high level, by Ghandi et al. 2023b and other LLM reasoning work. However, one of our main contributions is developing and conceptualizing a new complex and interactive testbed. Ghandi et al. 2023b used 2-player negotiation or matrix games, and their approach didn’t generalize to complex setups beyond 2 players. The authors also used few-shot demonstrations without an interactive setup. In contrast, we use completely autonomous multi-agent interactive negotiation. Besides, our negotiation game is substantially more complex than the “Deal or No Deal” dataset and has a semantically rich simulation. Our work is also the first to conceptualize attacks between agents and show how greedy/adversarial agents can indeed affect/manipulate the group.
>
> ### **3- Capabilities required for negotiation**
>
> Thank you for the helpful suggestion. We identified some missing capabilities in our paper. For example, GPT-3.5 showed *weaker arithmetic skills* (indicated by the percentage of deals violating the minimum threshold). GPT-3.5 also performed worse when asked to estimate the preferences of other agents (~40% vs ~60% in the case of GPT-4), which is related to *theory-of-mind* skills. Across one model (GPT-4), we showed in our ablation that *planning* improves the performance as well as explicitly instructing the model to infer *others’ preferences* before suggesting deals. We also showed that the performance decreases when the model does not iteratively *adapt* or react to previously suggested deals and instead makes proposals that only meet its score (even if they are less than its ideal deal). In our qualitative examples, we also identified a *lack of exploration*, which may explain some of the unsuccessful sessions of the game. We will restructure the experiments to highlight these points more and point to evidence for each of them.

---

> > ### Author Response · Authors · 2023-11-17
> > **New experiments**
> >
> > ### **4- Additional Experiments**
> >
> > Thank you for the great suggestions. We now add more experiments. Please refer to the supplementary material for figures.
> >
> > The mixed population is a plausible scenario that we originally considered but didn’t proceed with due to the lower performance of GPT-3.5 since it would naturally reduce the success rate. We now showcase some examples of it that support our original hypothesis. Since the game is based on cooperative bargaining and collaboration (as a baseline), it requires at least 5 consenting parties, including the two veto parties (i.e., the deal must satisfy their BATNAs). GPT-3.5 agents frequently violate their own BATNA rule, which might lead to an unsuccessful outcome for the whole group. For example, we now add an experiment where the leading agent is GPT-3.5. Even if it proposes a deal that satisfies the BATNA’s of all agents except itself, the game would still be unsuccessful for the entire group, which is what we observed in the new experiment (see Figure 1 in the new attached supplementary material pdf). The game’s success also decreases when two of the agents are GPT-3.5, which also might be explained by the inadequate arithmetic calculations of deals. Interestingly, even in the cooperative game, GPT-3.5 agents can get a lower score compared to their counter GPT-4 agents in the cooperative game (see Figure 2 in the supplementary material).
> >
> > [1] Abdelnabi et al. "More than you've asked for: A Comprehensive Analysis of Novel Prompt Injection Threats to Application-Integrated Large Language Models." AISec workshop, 2023.
> >
> > | Experiment                                                                             |  Success rate |
> > | ---------------------------------------------------------------------------------------- | ------- |
> > | All GPT-4 - All cooperative (originally reported in the paper) |  81% |
> > | All GPT-3 - All cooperative (originally reported in the paper)             | 20% |
> > | Mixed models (leading: GPT-3.5) - All cooperative (**new**)                          | 50% |
> > | Mixed models (two beneficiary agents are GPT-3.5) - All cooperative (**new**)  | 62% |
> >
> > - - -
> >
> > We also add one experiment where the leading agent $p_1$ is greedy. This significantly reduced the success rate since the leading agent is the agent that makes the final deal suggestion. This itself could potentially be an interesting attack vector where the leading agent is prompted indirectly via a third party [1] to be greedy, which would sabotage the negotiation for the entire group (see Figure 3 in the supplementary material). We also add an experiment where two agents are greedy (they are different from the greedy agent reported originally in the paper), which also reduces the success of the game and reduces the score of other cooperative agents (see Figure 4 in the supplementary material).
> >
> > | Experiment                                                                             |  Success rate |
> > | ---------------------------------------------------------------------------------------- | ------- |
> > | One greedy - all GPT-4 (originally reported in the paper) |  57 % |
> > | Leading is greedy - all GPT-4 (**new**) | 27% |
> > | Two beneficiary agents are greedy - all GPT-4 (**new**)  | 65% |
> >
> > - - -
> >
> > We originally excluded attacks between mixed populations in order to exclude confounding factors (note that the success decreases when some agents are GPT-3.5 due to their noisy/inconsistent deals even if they are incentivized to cooperate – therefore, the decrease in performance cannot entirely be attributed to the attack itself and how agents mounted/planned it). We show in Figure 5 in the supplementary material that the saboteur GPT-3.5 agent doesn’t follow the attack’s instructions (i.e., maximizing its score, minimizing the target’s score).
> >
> > | Experiment                                                                             |  Success rate |
> > | ---------------------------------------------------------------------------------------- | ------- |
> > |Target/saboteur are GPT-4  (originally reported in the paper) | 58% |
> > | Saboteur is GPT-3.5 (**new**) | 52% |

---

> > > ### Author Response · Authors · 2023-11-22
> > >
> > > We would also like to clarify that we didn't add an experiment with two saboteur agents because this would change the requirement for the deal to succeed compared to the baseline cooperative game and may make the comparison not straightforward. In the one-out game, we consider the negotiation successful if the rest of the 5 parties agree. This is the same condition in the cooperative game (at least 5 consenting parties). The two-out game would change this to 4 consenting parties.

---

> > > > ### Comment · Reviewer_f6ii · 2023-11-23
> > > >
> > > > Thank you for the clarifications, I especially appreciate additional experiments, and I'm more positive about this work. However, I also tend to agree with some of the concerns raised by other reviewers, in particular, those related to the novelty of this work, as well as the experimental testbed - my review outlines similar concerns. Given these concerns, I'm leaning to keep my score as it is, but will also consider raising it when making a final decision.

---

> ### Author Response · Authors · 2023-11-17
> **Additional questions**
>
> We answer the additional questions below.
>
> ### **1- Rational Agents**
>
> That is an interesting question. It is, however, not clear how to define rational agents in the scope of our considered game. According to [2], rational agents are defined as "A rational agent is one that acts so as to achieve the best outcome or, when there is uncertainty, the best expected outcome". It is unclear how the best outcome can be defined in our case (e.g., the maximum possible score for one agent, the maximum "feasible" score, or the maximum score for the group, etc.). It is also important to note that agents are not fed other agents' payoffs (i.e., they cannot directly observe the environment state). Therefore, quantifying the best outcome is not straightforward. It is also not a turn-taking game where there are clear transitions between states that determine the utility. LLMs are not inherently rational agents, beyond next-word prediction of the “best” token, as they have no direct reward function for these negotiation actions. We, however, think that it would be interesting to interpret their output based on the game’s reward and combine our approach with explicit and structured search and planning paradigms [3] that go beyond the CoT to create a search tree of feasible deals, simulate outcomes of actions (e.g., consequent actions of other agents and whether they would agree or not), and score deals accordingly. We leave these directions for future work.
>
> [2] Stuart Russell and Peter Norvig. "Artificial Intelligence: a modern approach".
>
> [3] Shibo Hao et al. "Reasoning with language model is planning with world model." arXiv preprint, 2023.
>
> ### **2- Attacks' setup**
>
> In the game "All in - one greedy"/"One out", are all the agents aware that there is one agent that is selfish/adversarial?
> No, agents are not fed the information that there are other greedy/adversarial agents. Notably, we observed that agents could sometimes identify the odd agent (e.g., other agents identified that the adversarial agent is considerably different than others and the greedy agent is insisting on some issues). We think that this could be an interesting follow-up defense, which we briefly outlined in the discussion, e.g., task a moderator to detect attacks. This could also potentially be a way to limit the adversary’s capabilities and force it to be subtle as an adaptive attack.
>
> ### **3- Hyperparameters**
>
> The choice of agents and options was mainly motivated by the base game found in negotiation training literature. When creating the new games, we experimented with a game that had a lower number of options and we observed that it usually had a higher success rate, possibly due to the lower number of possible combinations. This could generally be another option to modulate the difficulty of games (besides the one we already outlined in the paper), especially since it is possible to automatically rewrite the game by prompting the LLM to add/remove options.

---

> ### Author Response · Authors · 2023-11-23
>
> Thank you for your reply. We are glad to know that you found the additional experiments helpful. We would like to highlight again the novelty of our work regarding evaluating LLMs in a complex decision-making environment (as you pointed out in your original review, which was very encouraging for us to hear) with partial observations and a total of 720 possible deals with only a subset of them leading to success (a controllable hyperparameter that allows quantifying the performance and tuning the difficulty).

---

### Official Review · Reviewer_sx4h · 2023-11-03

**Soundness:** 3 good
**Presentation:** 3 good
**Contribution:** 3 good
**Rating:** 5
**Confidence:** 5

**Summary:**

This paper introduces negotiation games as an innovative evaluation benchmark for LLMs. These games assess LLMs' performance, limitations, and potential misuse in practical negotiation scenarios, such as customer service, contract agreements, and decision-making. The study demonstrates that GPT-4 significantly outperforms earlier models in negotiation tasks, showing strong zero-shot reasoning abilities. It explores agent interactions in unbalanced adversarial settings, revealing how agent behavior can be modulated to affect negotiation outcomes. The benchmark includes diverse negotiation games with multiple parties, different issues, and varying priorities, providing room for further enhancements. The paper plans to make its toolkit of negotiation games and code publicly available to facilitate future research in this area.

**Strengths:**

1. Practical Relevance: The choice of negotiation as an evaluation task is motivated by its practical importance in various real-life situations, such as customer service, contract agreements, and decision-making. LLMs are increasingly being used in such tasks, making their evaluation in negotiation crucial.

2. Interesting Adversarial Settings: The study explores agents' interactions in unbalanced adversarial settings, which are relevant for future autonomous systems with limited human oversight. It shows that agent behavior can be modulated to promote greediness or attack other agents, impacting negotiation outcomes.

3. Diverse Benchmark: The paper creates a diverse benchmark of negotiation games, including multiple parties, different issues, and varying priorities. This benchmark provides a quantifiable measure of LLM performance and room for further enhancements.

**Weaknesses:**

1. The negotiation games employed in this paper involve a simple setup with limited actions and a public communication channel. Real-world negotiations can be more complex, including private messages, alliances, and natural language conversations.

2. The study primarily considers adversarial players restricted by valid negotiation actions. Other forms of attacks, such as adversarial suffixes, are not explored.

3. While the paper highlights LLMs' strong zero-shot reasoning in negotiation games, it acknowledges that fine-tuning on real-world negotiation scenarios may be necessary for practical applications. Can this be improved?

4. Chain-of-Thought (CoT) prompting strategies are abductive in nature.  Employing that to improve reasoning is strange.

**Questions:**

The work of SocraSynth has received much attention, enjoying over 10k views.  Please articulate the differences between this work and the approach of SocraSynth, e.g., purposes, techniques, and applications.

For instance, as far as I can tell, SocraSynth focuses on knowledge synthesis and reasoning using LLMs, enabling the extraction of deep insights and information from these models. Negotiation games, on the other hand, assess LLMs' abilities in practical negotiation scenarios, emphasizing their interactive behavior and potential for manipulation.  Please comment on if this makes sense.

---

> ### Author Response · Authors · 2023-11-17
> **Thank you for your positive evaluation.**
>
> Thank you so much for your positive evaluation. We respond to the raised weaknesses below:
>
> ### **1- Games' Complexity**
> As discussed in the paper, our game is substantially more complex than previous work that evaluated LLMs on negotiation (e.g., Fu et al. [1]) with 2-players and one simple issue. Our work is the first to employ a *multi-agent* and *multi-issue* setup with *scorable negotiation* that clearly quantifies success and collaboration in role-play simulation where agents have real-world inspired goals. The game is an established exercise commonly used for teaching negotiation skills [2] and has been designed to balance success and collaboration. Our work is also the first to explore the *adversarial setup* in LLM negotiation in order to better match real-world scenarios that do not exclusively assume cooperation willingness. The targeted adversarial game shows agents forming *alliances* and coalitions, in which the leading agent agrees with the adversarial one against the target agent. By “limited action space”, we refer to the fact that agents in each turn either support previous deals or make new proposals. However, the total number of deals’ combinations is 720, out of them only 55 lead to an agreement, making the game have a large number of action space of deals at each round. We also would like to clarify that agents interact with *natural language* conversations; in addition to sharing deals suggestions, agents may emphasize the importance of some issues or indicate they are neutral and flexible about others.
>
> [1] Fu et al. "Improving language model negotiation with self-play and in-context learning from AI feedback." arXiv preprint, 2023
>
> [2] Susskind, Lawrence E. "Scorable games: A better way to teach negotiation."
>
> ### **2- Adversarial Setup**
>
> Restricting the game to valid negotiation actions is arguably a threat model that imposes higher constraints on the adversary, making it a more challenging setup that is also by far more technically interesting to study (e.g., the interaction dynamic between the attacker and other agents, how the attack plan and conduct the attacks). In addition, attacks like adversarial suffixes usually require white-box access to the model or a substitute model. Instead, we limit our threat model to black-box attacks. Our attacks can also be harder to detect than, e.g., jailbreaking and adversarial suffixes, and can be relevant as future work to study AI deception and manipulation [3,4] by providing a simulated environment.
>
> [3] Park et al. "AI deception: A survey of examples, risks, and potential solutions." arXiv preprint, 2023
>
> [4] Scheurer et al. “Technical Report: Large Language Models can Strategically Deceive their Users when Put Under Pressure”. arXiv preprint, 2023
>
> ### **3- Finetuning**
>
> Our work serves, among other things, as an easily adaptable benchmark for current and future models on an interactive multi-agent negotiation task that requires arithmetic, exploration, theory-of-mind, and planning capabilities. As we discussed in our paper, further fine-tuning might be required for practical applications. However, this is out of the current scope of the paper as fine-tuning models requires annotated data, possibly on human negotiation [5], which would need additional data collection considering our game. Additionally, access to GPT-4 for fine-tuning is not possible (at the time of submission). Our work may also contribute to creating fine-tuning data for smaller models by training on GPT-4 game sessions that reached successful collaboration. However, we leave these avenues for future work. It is also worth mentioning that our benchmark can be valuable in evaluating future/fine-tuned models that might have improved negotiation capabilities since the difficulty of the game can be easily adjusted by changing the agents’ scores.
>
> [5] Lewis et al. "Deal or no deal? End-to-end learning for negotiation dialogues." arXiv preprint, 2017

---

> > ### Author Response · Authors · 2023-11-17
> >
> > ### **4- CoT**
> > CoT has been shown to improve LLMs reasoning in many tasks, including arithmetic ones and commonsense reasoning [6], making it relevant to our task that spans these sub-tasks, among others. Decomposing the task into steps or intermediate sub-tasks (in our case, e.g., observing recent interactions, reasoning about others’ preferences, exploring next moves, etc.) has been used to improve problem-solving and reasoning via a consistent chain of thought [7,8]. It is arguable as well that our task partially falls under “abductive reasoning”. From [9]: “Abductive reasoning is a type of inference aiming at finding the minimal and most justified explanation for the set of phenomena or observations”. This applies to our case where agents only observe other agents’ actions and should reach conclusions about their preferences. CoT was also found to improve GPT-4’s performance on abductive reasoning [9].
> >
> > [6] Wei et al. "Chain-of-thought prompting elicits reasoning in large language models." NeurIPS, 2022.
> >
> > [7] Wu et al. "SPRING: GPT-4 Out-performs RL Algorithms by Studying Papers and Reasoning." arXiv preprint, 2023.
> >
> > [8] Gandhi et al. "Strategic Reasoning with Language Models." arXiv preprint, 2023.
> >
> > [9] Maksym Del and Mark Fishel. "True detective: a deep abductive reasoning benchmark undoable for GPT-3 and challenging for GPT-4." The 12th Joint Conference on Lexical and Computational Semantics. 2023.
> >
> > ### **Questions on SocraSynth**
> >
> > Thank you for pointing out this work. As you summarize, SocraSynth indeed aims to iteratively synthesize knowledge and extract different perspectives from LLMs via a debate between opposing and agreeing agents. While it is not directly related to negotiation and collaboration between agents to achieve a quantifiable task, we think it has an interesting conceptual resemblance to our work in terms of having a debate between agents and a refinement process to collaboratively reach a consensus (based on verifying facts and improving quality over generated arguments, not bargaining and compromising over partially shared preferences as in our work). An interesting future extension of our work would be to incorporate the debate idea within each party to reach better proposals. We will discuss this related work in the final version of the paper.

---

> > ### Comment · Reviewer_sx4h · 2023-11-22
> >
> > "Our work is also the first to explore the adversarial setup in LLM negotiation in order to better match real-world scenarios that do not exclusively assume cooperation willingness."
> >
> > The SocraSynth stands out as the first work of this area.  Please be impartial in acknowledgement.

---

> > > ### Author Response · Authors · 2023-11-22
> > >
> > > Thank you for pointing that out. We indeed did not want to make the impression of not acknowledging SocraSynth (http://infolab.stanford.edu/~echang/SocraSynth.html), and we added it to our related work discussion in the paper. We agree that SocraSynth studies an adversarial debate between agents. However, to the best of our knowledge, it is not studying negotiation, which clearly makes our work distinct. Our adversarial setup involves attacks between agents to, e.g., increase the payoff of the attacker, which is also conceptually different than the adversarial setup to synthesize knowledge.

---

### Public Comment · ~Guohao_Li1 · 2023-11-14
**Missing related work**

LLM-Deliberation explores the use of negotiation games as a benchmark for evaluating Large Language Models (LLMs). These text-based multi-agent, multi-issue negotiation games are designed to assess LLMs' reasoning and decision-making capabilities. The study specifically focuses on zero-shot Chain-of-Thought (CoT) prompting strategies and measures performance using various metrics, noting significant performance gaps between GPT-4 and earlier models. The paper also examines the generalization of these models to new games and setups, and their interaction dynamics in scenarios involving greedy and adversarial players.

Thanks for the great work. However, it could also be beneficial to discuss prior work on multi-LLM agents for the study of cooperative settings [1].

[1] Li, Guohao, Hasan Abed Al Kader Hammoud, Hani Itani, Dmitrii Khizbullin, and Bernard Ghanem. "CAMEL: Communicative Agents for" Mind" Exploration of Large Language Model Society." NeurIPS 2023

---

> ### Author Response · Authors · 2023-11-21
>
> Thank you for the suggestion. We now briefly discuss previous work collaboration/debate between agents in order to achieve tasks or synthesize knowledge and point out the differences to our setup of negotiation.

---

### Author Response · Authors · 2023-11-17

We thank the reviewers for their feedback and for their positive evaluation of our paper’s writing and contributions regarding its practical relevance, the interesting adversarial setting, its proposed diverse benchmark, and its novel setup and results on the utility of LLMs in complex decision-making settings. We address reviewers’ concerns in our replies below. We believe that reviewers’ comments are addressable with additional clarifications regarding our contributions and reorganizing the corresponding parts in the paper to highlight them. **We thank the reviewers for raising interesting discussion items and suggestions regarding additional experiments, which we currently added**. We briefly discuss the new results in our replies and we also attach in the supplementary material additional figures (in a PDF file) and the prompts of games. We are happy to answer any more questions the reviewers might have.

---

### Author Response · Authors · 2023-11-21
**Paper is updated.**

Dear AC and reviewers,

We would like to let you know that we have now also updated our paper in order to incorporate the feedback and new experiments we added.

A summary of changes:
- Restructuring the related work
- Adding an analysis of needed skills and capabilities in our experiments.
- Adding the new experiments on the mixed population of models and variations of attack experiments.
- Updating the supplementary material in the paper to include the new results, and our analysis to show that GPT-4 models are consistent with their payoffs (e.g., regarding issues such as environmental protection).

We kindly request you to please share your thoughts about our paper and to consider increasing the rating if you agree with our response. We are happy to answer further questions.

---

### Meta-Review · Area_Chair_tgUS · 2023-12-10

**Metareview:**

This paper is investigating LLM agents in multi-agent negotiation settings. The main contribution of the paper is providing a framework/testbed for evaluating LLMs in multi-agent settings through scorable negotiation games. I believe the contribution of the paper is certainly relevant to ICLR community as the LLMs as decision making agents is a timely topic. However, while the paper is conceptually strong and proposed testbed has clear merits, some reviewers and I have found the technical merit to be weak. In essence, paper is based on prompting LLMs, however, it does not provide a new algorithm or approach to the issue of multi-agent games. Additionally, some of the insights are not really surprising in light of the prior literature (such as GPT4 having a much better performance). For a future version, the authors could benefit from incorporating reviewer suggestions. I also suggest they could emphasize the unique decision making aspect of this setting. For instance, they could explore the impact of fine-tuning or planning-oriented prompted techniques such as tree/graph-of-thought.

**Justification For Why Not Higher Score:**

N/A

**Justification For Why Not Lower Score:**

N/A

---

### Decision · Program_Chairs · 2024-01-16

Reject